# Trends and projections of the global burden of T2DM-associated CKD related to high BMI: A global burden of disease study 2021

Jing Zhang[1]☉, Wenxuan Li[1]☉, Zhen Sun[2], Yunyang Wang[1], Yu Xue[1], Ke Si[1], Yajing Huang[1], Wenshan Lv[1], Lili Xu[1], Yangang Wang☉[1]*

1 Department of Endocrinology and Metabolism, The Affiliated Hospital of Qingdao University, Qingdao, China, 2 Department of Neurology, The Affiliated Hospital of Qingdao University, Qingdao, China

☉ These authors contributed equally to this work.
* wangyg@qdu.edu.cn

## Abstract

### Background

Type 2 diabetic-associated chronic kidney disease (T2DM-Associated CKD), a leading cause of end-stage renal disease, is exacerbated by rising particularly high body mass index (BMI) rates. This study examines the global burden of T2DM-Associated CKD attributable to high BMI from 1990 to 2021 and projects future trends using the Global Burden of Disease (GBD) 2021 data.

### Methods

GBD 2021 data from 204 countries were analyzed to assess mortality, disability-adjusted life years (DALYs) and corresponding age-standardized rates of T2DM-Associated CKD linked to high BMI. Bayesian Age-Period-Cohort modeling was used for projections, with stratification by age, gender, and Socio-Demographic Index (SDI). Statistical analyses were conducted using R software.

### Results

In 2021, high BMI-related T2DM-Associated CKD caused 173,263 deaths and 4.3 million DALYs. Age-standardized rates declined globally but showed regional disparities, with Andean Latin America having the highest burden. Women had higher absolute burdens, while men showed higher standardized rates. Projections indicate continued increases in mortality and DALY rates through 2050. Emerging therapies, such as GLP-1 receptor agonists (RAs) and SGLT2 inhibitors (SGLT2i), could potentially alter these trends, especially in high-risk regions.

**Data availability statement:** The dataset supporting the conclusions of this article is available in the Global Burden of Disease (GBD) repository, https://vizhub.healthdata.org/gbd-results/.

**Funding:** The author(s) received no specific funding for this work.

**Competing interests:** The authors have declared that no competing interests exist.

## Conclusions

High BMI significantly drives the T2DM-Associated CKD burden, necessitating targeted overweight/obesity prevention and improved healthcare access, particularly in high-risk regions. Monitoring trends is crucial for effective interventions.

## Introduction

In recent years, type 2 diabetes (T2D) and its associated complications, particularly type 2 diabetic-associated chronic kidney disease (T2DM-Associated CKD), have emerged as major global health concerns. T2DM-Associated CKD remains one of the most common and serious complications of T2D, contributing significantly to the rising burden of end-stage renal disease (ESRD). According to cohort studies on diabetes in European and American countries, it is estimated that more than half of T2D patients may develop T2DM-Associated CKD [1]. T2DM-Associated CKD causes more than 50% of all ESRD [2], placing a heavy burden on socio-economic and medical resources.

Meanwhile, the prevalence of overweight/obesity continues to rise worldwide and has become a significant public health issue [3]. For example, the United States, one of the countries with the highest prevalence of overweight/obesity and overweight, is expected to see the prevalence increase to nearly 50% and 30%, respectively, by 2030. It is estimated that the total direct healthcare and indirect costs of obese and overweight individuals in the United States in 2016 were $1.72 trillion [4]. Numerous studies have shown that a high BMI increases the risk of diabetic nephropathy (DN) [5,6]. A meta-analysis of observational studies showed that the risk of DN increased by 16% for every 5 kg/m2 increase in BMI [7]. Additionally, a Mendelian Randomization study showed a significant association between an increase of one standard deviation in BMI and a 2.76-fold increase in the risk of DN [5].

Recently, Tan et al. conducted a comprehensive analysis of the global burden of CKD attributable to high BMI from 1990 to 2021, revealing a significant increase in CKD mortality and disability-adjusted life years (DALYs). They also predicted that the burden of CKD will continue to grow by 2035, particularly in low- and middle-income countries [8]. In contrast, our study extends the projection horizon to 2050, offers a more granular stratification by age, sex, and region. By integrating data from GBD 2021, we aim to provide insights into potential public health interventions to address this growing challenge.

## Methods

### Study design and data sources

This study utilized data from the GBD 2021 study, which contains comprehensive data from 204 countries and regions. It covers 369 diseases and injuries, providing indicators such as prevalence, mortality, and DALYs at the global, regional, and national levels, as well as a comparative analysis of 87 risk factors [9]. The dataset covers the period from 1 January 1990–31 December 2021. The GBD data come

from population censuses, household surveys, civil registration and vital statistics, disease notifications, disease registries, health service use, air pollution monitoring, satellite images and other sources [10]. These data sources provide reliable estimates of the burden of disease using standardized methods across regions and time periods.

We selected individuals aged 25 to ≥95, including males, females, and both sexes combined, to assess the global burden of disease attributed to high BMI-related T2DM-Associated CKD in 204 countries and territories from 1990 to 2021. The analysis included data on the number of deaths, DALYs, and their corresponding age-standardized rates (per 100,000 people), ensuring comprehensive coverage of trend analysis. In this study, T2DM-associated CKD was attributed based on GBD 2021 data. It should be noted that the GBD database does not provide indicators such as proteinuria, eGFR, or KDIGO staging to distinguish specific clinical phenotypes. Therefore, we used the term "T2DM-associated CKD" to avoid overinterpretation of clinical phenotypes. GBD data can be obtained from https://vizhub.healthdata.org/gbd-results/.

## Definitions

For adults aged 20 and above, a high BMI is defined as a BMI of more than 25 kg/m². DALYs represent the sum of years of life lost due to premature mortality and disability, and a DALY can be regarded as all healthy life years lost. It is currently the most widely used and representative indicator for assessing and measuring the overall burden of disease [11]. The calculation of the uncertainty interval (UI) is based on 1000 sample estimates for each parameter. The range of the 95% UI is determined by the 25th and 975th values of the 1000 sorted estimates, which are used as the lower and upper limits of the interval, respectively. The Socio-demographic Index (SDI) is an index proposed in the World Development Report 2015 to reflect the development status of a geographical location. It is calculated based on the lagged distribution of per capita income, the average education level of the population aged 15 and above, and the total fertility rate of the population under 25 [9]. The 204 countries and territories are divided into five groups according to their SDI quintiles: high SDI (>0.805126), high-middle SDI (0.689504–0.805126), middle SDI (0.607679–0.689504), low-middle SDI (0.454743–0.607679) and low SDI (≤0.454743).

## Statistical analysis

**Joinpoint regression analysis.** We performed regression analyses using Joinpoint software (version 5.4.0; National Cancer Institute) to identify key turning points in trends in the burden of disease. Joinpoint regression analyses identify best-fit points, or 'inflection points,' by dividing the data into line segments and fitting regressions to each segment, and assesses the change in trend between the inflection points. An annual percentage change (APC) was calculated for each line segment, indicating the annual rate of change over a given time period. By weighting and averaging the APC of each segment, the estimation yields the average annual percentage change (AAPC), which reflects the overall trend of the entire time series. Statistical significance was tested by Monte Carlo permutation (MC permutation), and a p-value of <0.05 was considered statistically significant [12].

**Decomposition analysis.** This study used decomposition analysis to assess the relative contributions of ageing, population, and epidemiological changes to the global burden of T2DM-Associated CKD attributable to high BMI. Ageing of the population was assessed by analyzing changes in the age structure – the effect of changes in the proportion of the elderly population on the burden of disease. Population was determined by changes in the overall size of the population. Epidemiological changes refer to changes in the incidence of T2DM-Associated CKD related to high BMI, which are independent of population dynamics [13].

**Bayesian Age-Period-Cohort (BAPC).** We used the BAPC model to predict ASMR and ASDR from 2022 to 2050. The core idea is to use a Bayesian method to fit historical data to obtain the past distributions of age, period, and cohort effects, and then use Bayesian inference to extrapolate future disease burden trends using current data [14].

All analyses and graphical visualizations were performed using R software (version 4.3.0). P<0.05 was considered statistically significant.

## Ethical consideration

This study used publicly available data from the GBD study and therefore did not require ethical approval. No human participants were directly involved. All methods were performed in accordance with the relevant guidelines and regulations.

## Results

In 2021, there were 173,263 deaths from T2DM-Associated CKD worldwide related to high BMI, with an ASMR of 2.07 per 100,000, an increase of 78.4% since 1990 (Table 1). Meanwhile, the number of DALYs has increased by 3125105 over the past 32 years, with an ASDR of 50.14 per 100,000 population in 2021 (Table 2). Among the 21 GBD regions, the highest T2DM-Associated CKD ASMR and ASDR associated with high BMI were in Andean Latin America (6.9 and 147.4 per 100,000 population, respectively), followed by Central Latin America (5.06 and 125.48 per 100,000 population, respectively) and the Caribbean (4.85 and 112.51 per 100,000 population, respectively). The regions with the lowest ASMR and ASDR are Eastern Europe (0.51 per 100,000 population) and Australasia (15.98 per 100,000 population). In terms of EAPC, between 1990 and 2021, ASMR EAPC and ASDR EAPC were positive everywhere except Central Europe, High-income Asia Pacific and Southern Latin America, implying that the burden of disease due to high BMI is increasing in the vast majority of regions. (Tables 1 and 2).

Table 1. Deaths and ASMR of T2DM-Associated CKD attributable to high BMI in 1990 and 2021 and the temporal trends from 1990 to 2021.

| Death location | 1990 | | 2021 | | 1990–2021 |
| --- | --- | --- | --- | --- | --- |
| | Deaths (95% UI) | ASMR per 100,000 (95% UI) | Deaths (95% UI) | ASMR per 100,000 (95% UI) | EAPC in ASMR (95% CI) |
| Southeast Asia | 1833 (646, 3904) | 0.79 (0.28, 1.68) | 10263 (3643, 22044) | 1.67 (0.59, 3.62) | 2.53 (2.48, 2.58) |
| East Asia | 8853 (3190, 18825) | 1.33 (0.49, 2.77) | 39725 (16343, 73073) | 2 (0.81, 3.67) | 1.26 (1.12, 1.39) |
| Global | 40479 (17726, 67320) | 1.16 (0.51, 1.92) | 173263 (76311, 288454) | 2.07 (0.91, 3.47) | 2.02 (1.94, 2.11) |
| Oceania | 74 (30, 136) | 2.98 (1.18, 5.7) | 287 (113, 506) | 4.57 (1.76, 8.27) | 1.27 (1.13, 1.4) |
| Central Asia | 179 (86, 261) | 0.39 (0.19, 0.59) | 720 (374, 1085) | 0.94 (0.48, 1.45) | 2.28 (1.77, 2.8) |
| Central Europe | 931 (469, 1336) | 0.66 (0.33, 0.94) | 1480 (732, 2376) | 0.62 (0.31, 0.99) | −0.23 (−0.47, 0) |
| Eastern Europe | 542 (256, 797) | 0.2 (0.09, 0.29) | 1859 (948, 2603) | 0.51 (0.26, 0.71) | 3.07 (2.66, 3.48) |
| High-income Asia Pacific | 2197 (1028, 3989) | 1.24 (0.57, 2.27) | 7347 (3055, 13133) | 1.13 (0.49, 1.98) | −0.25 (−0.38, −0.13) |
| Australasia | 69 (36, 99) | 0.32 (0.16, 0.45) | 325 (161, 477) | 0.53 (0.27, 0.78) | 2.48 (1.98, 2.98) |
| Western Europe | 4453 (2041, 6873) | 0.74 (0.34, 1.13) | 11387 (4914, 18310) | 0.91 (0.39, 1.41) | 1.1 (0.88, 1.32) |
| Southern Latin America | 1313 (603, 1978) | 3 (1.37, 4.52) | 2551 (1173, 3911) | 2.81 (1.29, 4.26) | 0.08 (−0.29, 0.45) |
| High-income North America | 4667 (2251, 6961) | 1.29 (0.62, 1.88) | 28244 (12799, 45533) | 4.05 (1.86, 6.4) | 4.02 (3.77, 4.27) |
| Caribbean | 619 (257, 1060) | 2.57 (1.07, 4.42) | 2643 (1100, 4369) | 4.85 (2.02, 7.98) | 2.74 (2.52, 2.96) |
| Andean Latin America | 812 (306, 1336) | 4.38 (1.64, 7.27) | 3938 (1682, 6305) | 6.9 (2.93, 11.19) | 1.41 (1.13, 1.7) |
| Central Latin America | 1911 (843, 3178) | 2.65 (1.14, 4.57) | 12418 (5868, 19475) | 5.06 (2.39, 8) | 2.63 (2.11, 3.15) |
| Tropical Latin America | 2090 (873, 3375) | 2.63 (1.09, 4.4) | 9399 (4315, 14026) | 3.77 (1.72, 5.68) | 1.14 (0.93, 1.35) |
| North Africa and Middle East | 4419 (1942, 6922) | 3.12 (1.31, 5) | 16440 (7457, 25621) | 4.17 (1.85, 6.63) | 1.01 (0.89, 1.12) |
| South Asia | 2914 (1216, 5660) | 0.56 (0.24, 1.13) | 15754 (6786, 28269) | 1.12 (0.48, 2.03) | 2.27 (2.19, 2.35) |
| Central Sub-Saharan Africa | 366 (137, 710) | 1.97 (0.72, 3.72) | 1270 (467, 2385) | 2.92 (1.04, 5.64) | 1.07 (0.95, 1.18) |
| Eastern Sub-Saharan Africa | 906 (314, 1963) | 1.41 (0.5, 3.1) | 3135 (1153, 5897) | 2.26 (0.83, 4.37) | 1.39 (1.32, 1.46) |
| Southern Sub-Saharan Africa | 245 (95, 402) | 1.02 (0.38, 1.74) | 936 (365, 1567) | 1.89 (0.73, 3.34) | 2.5 (2.18, 2.83) |
| Western Sub-Saharan Africa | 1086 (388, 1971) | 1.46 (0.52, 2.79) | 3141 (1377, 5301) | 1.97 (0.82, 3.47) | 0.7 (0.56, 0.85) |

**Table 2. DALYs and ASDR of T2DM-Associated CKD attributable to high BMI in 1990 and 2021 and the temporal trends from 1990 to 2021.**

| DALY | 1990 | | 2021 | | 1990–2021 |
|---|---|---|---|---|---|
| Location | DALYs (95% UI) | ASDR per 100,000 (95% UI) | DALYs (95% UI) | ASDR per 100,000 (95% UI) | EAPC in ASDR (95% CI) |
| Global | 1197972 (539562, 1925808) | 30.82 (13.87, 49.54) | 4323077 (1942948, 6820314) | 50.14 (22.56, 79.15) | 1.68 (1.61, 1.76) |
| Southeast Asia | 58753 (20349, 126543) | 21.99 (7.69, 47.03) | 306887 (110217, 640064) | 44.54 (15.93, 93.97) | 2.45 (2.37, 2.53) |
| East Asia | 261721 (93741, 564170) | 31.37 (11.36, 66.67) | 993095 (395037, 1787618) | 45.95 (18.54, 82.57) | 1.26 (1.12, 1.4) |
| Oceania | 2439 (998, 4377) | 77.67 (31.42, 141.79) | 8561 (3387, 15057) | 109.87 (43.34, 192.6) | 1.03 (0.9, 1.17) |
| Central Asia | 13048 (6594, 17962) | 27.74 (13.97, 38.46) | 33078 (18149, 45984) | 39.17 (21.27, 55.11) | 0.86 (0.58, 1.14) |
| Central Europe | 32716 (17770, 44575) | 21.97 (11.97, 29.66) | 43859 (22920, 63967) | 19.72 (10.44, 28.06) | −0.32 (−0.42, −0.21) |
| Eastern Europe | 38076 (19365, 52454) | 13.74 (6.86, 18.81) | 65939 (36890, 87680) | 18.41 (10.32, 24.6) | 0.79 (0.69, 0.89) |
| High-income Asia Pacific | 53801 (25195, 92291) | 27.53 (12.7, 47.96) | 127359 (55706, 222942) | 25.17 (11.56, 43.11) | −0.15 (−0.27, −0.04) |
| Australasia | 2803 (1462, 3827) | 12.15 (6.35, 16.47) | 8665 (4701, 11963) | 15.98 (8.7, 21.88) | 1.13 (0.86, 1.4) |
| Western Europe | 125066 (56308, 182818) | 21.34 (9.73, 30.8) | 220490 (99501, 327778) | 21.57 (9.99, 31.71) | 0.17 (0.08, 0.27) |
| Southern Latin America | 30034 (14170, 43185) | 65.49 (30.74, 93.96) | 51699 (25085, 74471) | 58.73 (28.84, 84.13) | −0.09 (−0.41, 0.23) |
| Caribbean | 15970 (7014, 25427) | 62.07 (27.05, 99.53) | 60834 (26524, 93605) | 112.51 (49.26, 172.98) | 2.55 (2.36, 2.73) |
| High-income North America | 133408 (66239, 185639) | 38.61 (19.45, 53.55) | 607345 (285454, 911077) | 94.03 (44.88, 138.06) | 3.16 (2.94, 3.39) |
| Andean Latin America | 19405 (7651, 29693) | 96.96 (37.64, 151.31) | 86513 (38663, 133617) | 147.4 (65.4, 228.82) | 1.3 (1.03, 1.57) |
| Central Latin America | 53959 (25535, 83738) | 65.91 (30.45, 104.61) | 318450 (159394, 480567) | 125.48 (62.43, 190.67) | 2.52 (2.02, 3.02) |
| North Africa and Middle East | 119890 (56801, 179674) | 72.24 (33.46, 109) | 423076 (210977, 631071) | 93.06 (44.34, 140.97) | 0.86 (0.82, 0.91) |
| Tropical Latin America | 60216 (26267, 89549) | 66.26 (28.34, 100.9) | 219886 (104654, 313390) | 85.51 (40.51, 122.33) | 0.69 (0.48, 0.91) |
| South Asia | 98184 (39673, 186530) | 16.29 (6.6, 31.86) | 496214 (217181, 861080) | 32.31 (14.05, 56.5) | 2.35 (2.27, 2.42) |
| Central Sub-Saharan Africa | 11116 (4463, 21206) | 49.23 (19.26, 94.05) | 38651 (15190, 69360) | 70.05 (26.82, 129.83) | 0.95 (0.86, 1.05) |
| Eastern Sub-Saharan Africa | 24353 (8486, 51559) | 32.76 (11.45, 70.01) | 82433 (31225, 148717) | 49.93 (18.52, 92.38) | 1.2 (1.14, 1.27) |
| Southern Sub-Saharan Africa | 9270 (4178, 13819) | 33.62 (14.58, 50.72) | 30524 (14275, 48108) | 52.65 (23.54, 84.77) | 1.69 (1.46, 1.92) |
| Western Sub-Saharan Africa | 33744 (13091, 57151) | 38.94 (14.72, 67.27) | 99519 (48176, 152321) | 50.51 (23.27, 80.56) | 0.67 (0.57, 0.77) |

Between 1990 and 2021, the number of deaths and DALYs associated with T2DM-Associated CKD related to high BMI and their corresponding age-standardized rates showed a continuous growth trend worldwide. The data shows that the disease burden in women is consistently higher than in men, and the gender gap has gradually widened over time. However, when we look at ASMR and ASDR, we find that the burden on men is consistently higher than that on women, and the gender gap is widening year by year. (Fig 1)

Among the 204 countries and territories, American Samoa had the highest T2DM-Associated CKD ASMR and ASDR related to high BMI in 2021, at 26 and 565.86 per 100,000, respectively. This was followed by the Northern Mariana Islands (20.31 and 434.07 per 100,000, respectively) and Nauru (18.49 and 453.6 per 100,000, respectively). In contrast,

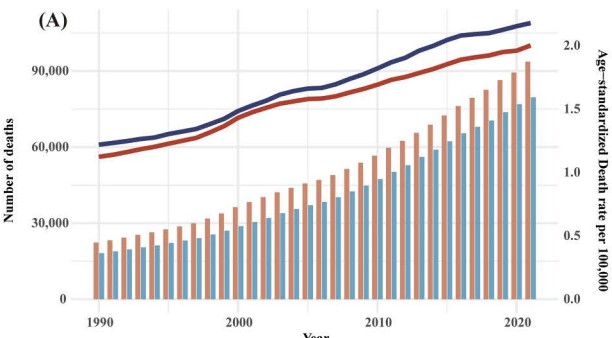
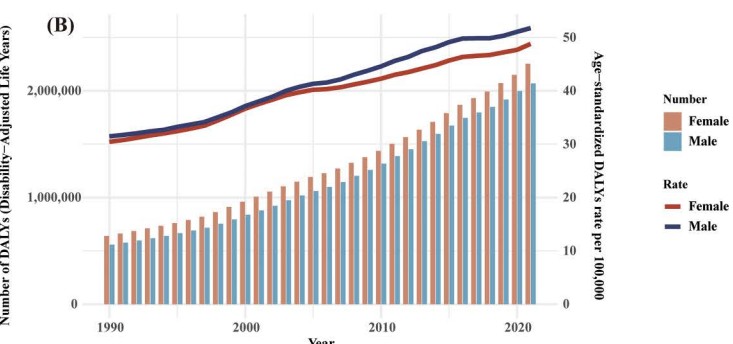

**Fig 1. Trends in global Deaths and DALYs and corresponding age-standardized rates of T2DM-Associated CKD attributable to high BMI, 1990-2021. (A)** Trends in deaths and ASMR; and **(B)** Trends in DALYs and ASDR. BMI, body mass index; DALYs, disability-adjusted life years; ASMR, Age-standardized mortality rate; ASDR, Age-standardized disability-adjusted life year rate.

the countries with the lowest ASMR and ASDR were Ukraine (0.13 per 100,000) and Turkana (10.79 per 100,000). (Tables in S1 and S2 Tables)

In 2021, ASMR and ASDR as a whole showed a downward trend with increasing SDI, indicating that the higher the level of socioeconomic development, the lower the ASMR and ASDR of T2DM-Associated CKD caused by high BMI. However, this trend is not linear. In areas with an SDI < 0.7, ASMR and ASDR actually increase with increasing SDI, which may be related to the gradual improvement of medical resources and the increase in disease diagnosis rates in these areas. In addition, lifestyle changes in low SDI areas during the early stages of economic development, such as the westernization of the diet and a reduction in physical activity, may also lead to an increase in the disease burden.

Over the past 32 years, the global ASMR and ASDR associated with high BMI-related T2DM-Associated CKD have generally decreased with increasing SDI. However, there are significant regional differences, mainly reflected in the fact that ASMR and ASDR have increased in regions with low SDI, while they have mostly remained stable or decreased in regions with high SDI. (Fig 2)

Both ASMR and ASDR are expected to continue increasing between 1990 and 2021, both globally and in the five SDI regions. ASMR is expected to increase the most in the high SDI region and the least in the low SDI region. ASDR is expected to increase the most in the middle SDI region and the least in the low SDI region. In 2021, ASMR and ASDR are highest in the middle SDI region (2.46 and 58.59 per 100,000, respectively) and lowest in the low SDI region (1.50 and 38.01 per 100,000, respectively). In addition, the ASMR and ASDR in the middle SDI region have been the highest for 32 years. In summary, although the ASMR and ASDR have increased in different regions worldwide, the differences between the SDI regions are still significant. (Fig 3)

From 1990 to 2021, the age-standardized burden of T2DM-Associated CKD related to high BMI showed a significant upward trend. The plot of the joinpoint regression model for the global ASMR of T2DM-Associated CKD related to high BMI from 1990 to 2021 contains five inflection points and is divided into six segments: 1990–1997, APC = 1.64%; 1997–2000, APC = 2.84%; 2003–2007, APC = 1.01%; 2007–2015, APC = 2.13%; 2015–2021, APC = 1.11%. Male ASMR increased annually by an average of 1.90%, i.e., AAPC = 1.90 (95% CI, 1.79–2.0; P < 0.05), and female ASMR had an AAPC = 1.87 (95% CI, 1.75–2.0; P < 0.05). Overall, the AAPC for the global ASMR of T2DM-Associated CKD related to high BMI from 1990 to 2021 was 1.90 (95% CI, 1.78–2.02; P < 0.05).(additional file 1) The ASDR joinpoint regression model chart contains five turning points and is divided into six sections: 1990–1996, APC = 1.19%; 1996–2003, APC = 2.64%; 2003–2007, APC = 1.03%; 2007–2016, APC = 1.66%; 2016–2019, APC = 0.40%; 2019–2021, APC = 1.65%. The AAPC for males, females and both sexes were 1.63 (95% CI, 1.54–1.73, P < 0.05), 1.53 (95% CI, 1.43–1.63, P < 0.05), and 1.58 (95% CI, 1.49–1.68, P < 0.05), respectively. (Fig 4).

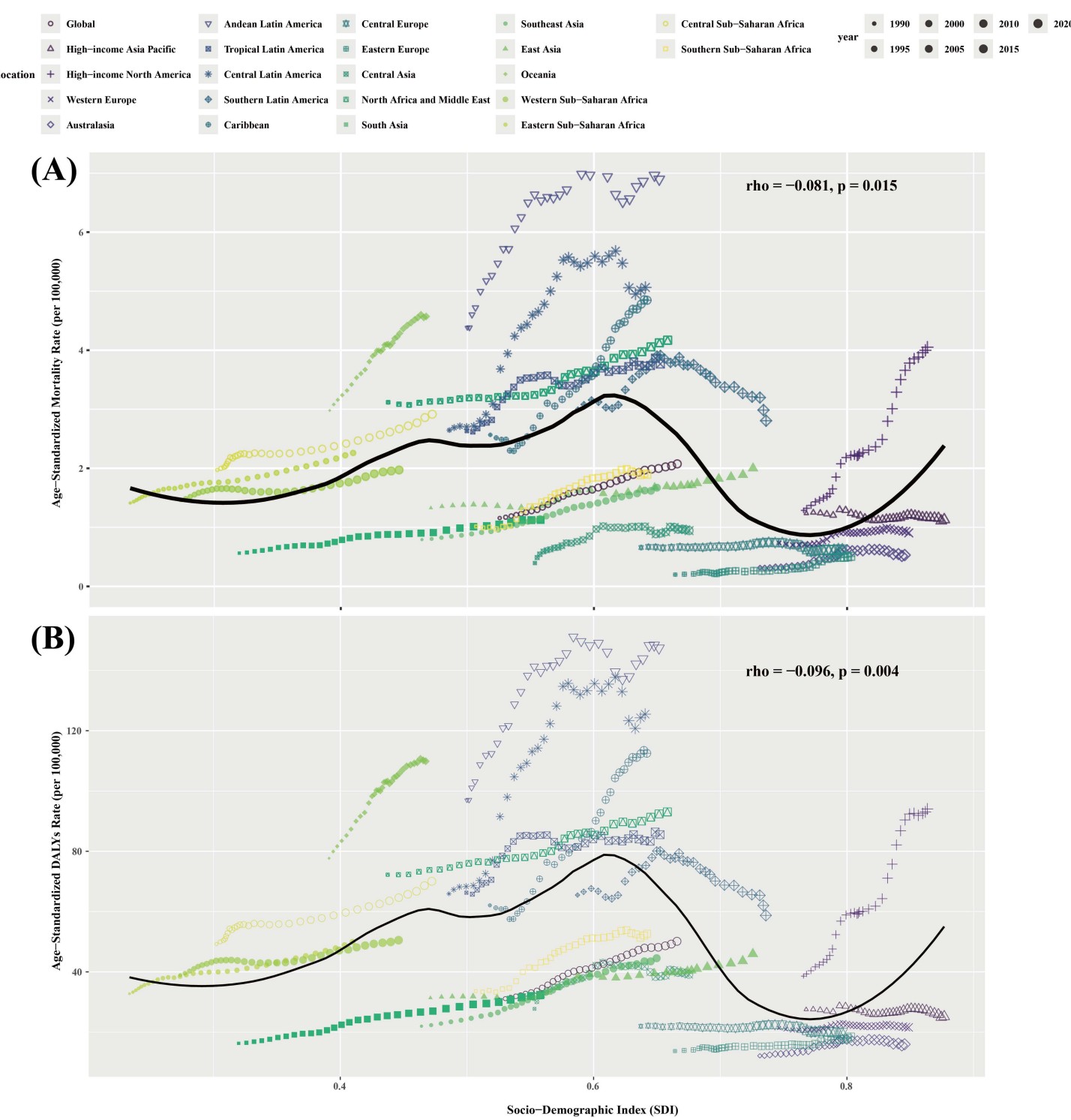

**Fig 2. ASMR and ASDR of T2DM-Associated CKD attributable to high BMI across 21 GBD regions by the socio-demographic index for both sexes combined, 1990–2019. (A)** ASMR; and **(B)** ASDR. BMI, body mass index; ASMR, Age-standardized mortality rate; ASDR, Age-standardized disability-adjusted life year rate.

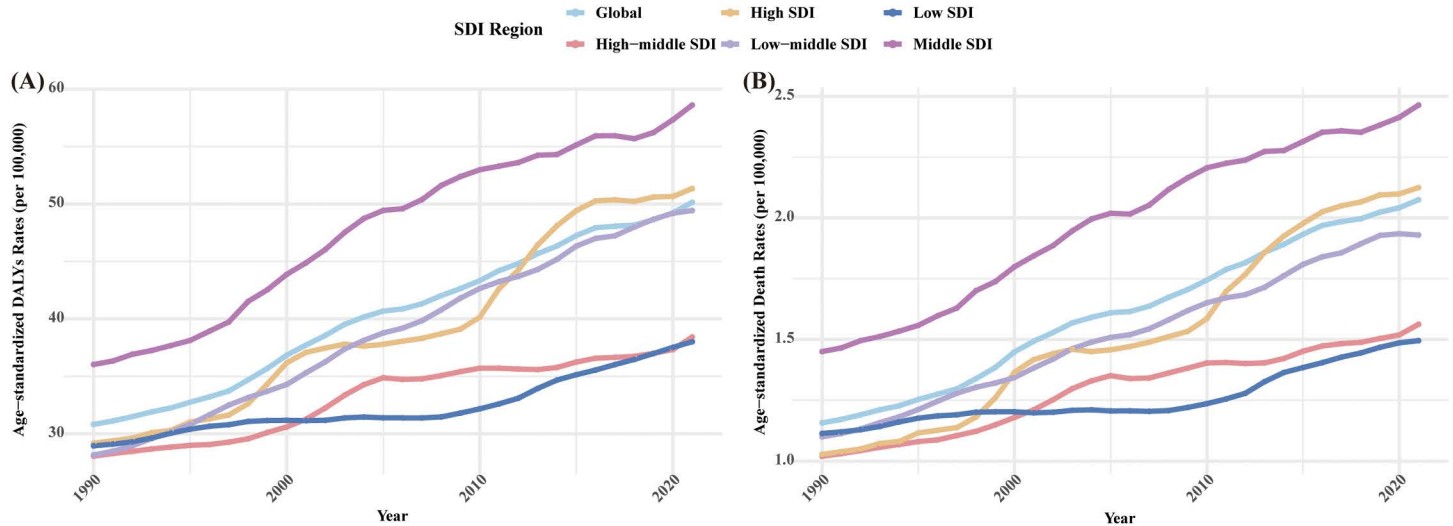

**Fig 3. Changes in the ASMR and ASDR of T2DM-Associated CKD attributable to high BMI globally and in different SDI regions from 1990 to 2021. (A)** ASMR; and **(B)** ASDR. BMI, body mass index; ASMR, Age-standardized mortality rate; ASDR, Age-standardized disability-adjusted life year rate. SDI, socio-demographic index.

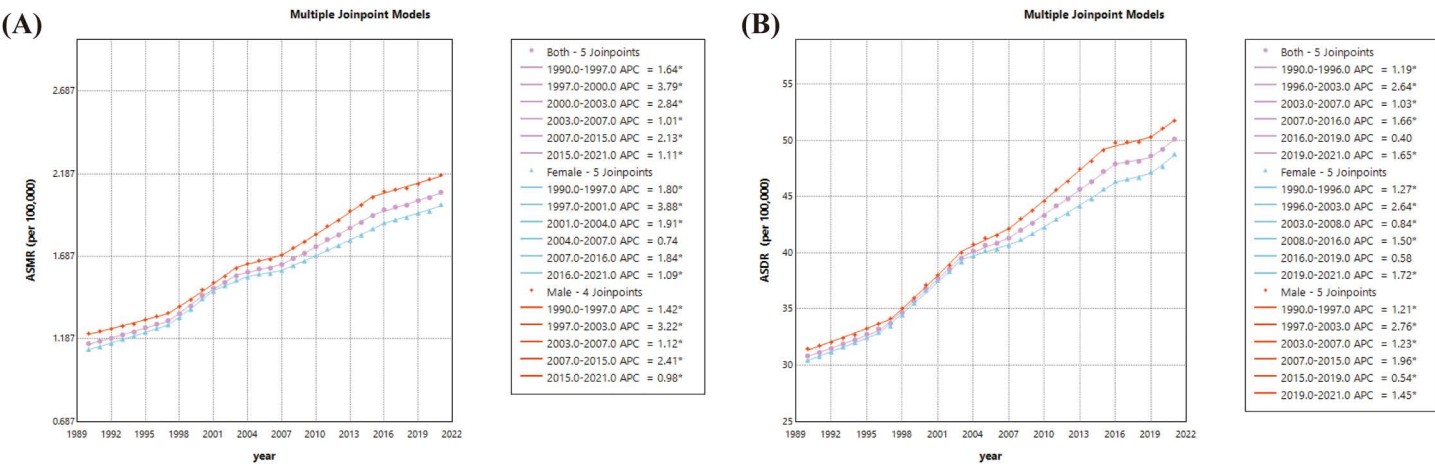

**Fig 4. Changes in time trends of joinpoint regression of T2DM-Associated CKD attributable to high BMI, 1990–2021. (A)** Time trends in ASMR; and **(B)** Time trends in ASMR. BMI, body mass index; ASMR, Age-standardized mortality rate; ASDR, Age-standardized disability-adjusted life year rate.

Fig 5 shows the factor decomposition of the changes in ASMR and ASDR for T2DM-Associated CKD related to high BMI from 1990 to 2021 globally. For ASMR, epidemiological changes are the most important driver, followed by population, and ageing has the smallest effect. Similar to ageing, epidemiological changes affect women more than men, while population affects men and women similarly. For ASDR, epidemiological changes remain the dominant factor driving overall growth for both sexes, followed by population. In contrast, ageing has a negative effect on the increase in ASDR, although this protective effect is minimal. (Fig 5).

Over the next 26 years, the ASMR and ASDR for both males and females are expected to rise steadily. By 2050, the ASMR of T2DM-Associated CKD related to high BMI in men is expected to increase to 5.21 per 100,000, an increase of

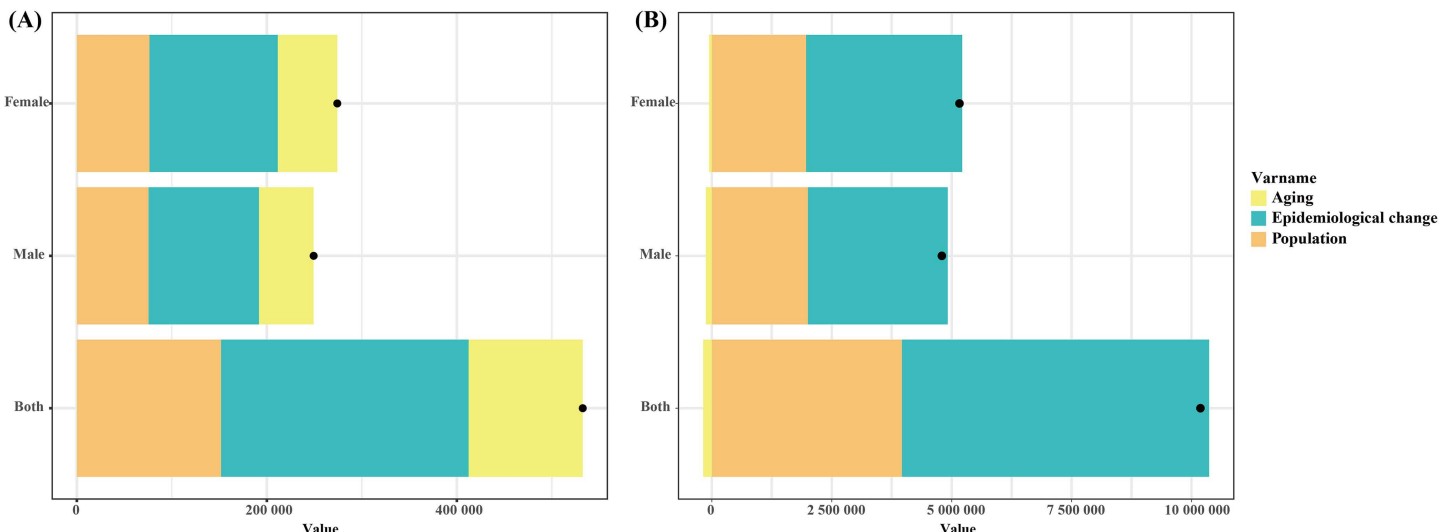

**Fig 5. Decomposition analysis of changes in ASMR and ASDR attributable to high BMI in global T2DM-Associated CKD from 1990 to 2021, by aging (yellow), epidemiological changes (green), and population growth (orange). The black dots representing the sum of the effects of these factors. (A)** Decomposition of changes in ASMR for both sexes, males, and females. **(B)** Decomposition of changes in ASDR for both sexes, males, and females. BMI, body mass index; ASMR, Age-standardized mortality rate; ASDR, Age-standardized disability-adjusted life year rate.

30.90% compared to 3.98 per 100,000 in 2021; the ASMR in women will increase from 3.66 per 100,000 in 2021 to 4.90 per 100,000, an increase of 33.88%. In addition, the ASDR for men will increase from 0.00094 per 100,000 in 2021 to 0.00127 per 100,000 in 2050, while for women it will increase from 0.00089 per 100,000 to 0.00124 per 100,000. These changes indicate that the impact of BMI on the disease burden of T2DM-Associated CKD will further increase in the future, and there is a significant upward trend for both men and women. (Fig 6).

## Discussion

This study included 204 countries and regions, assessed the global burden pattern of T2DM-Associated CKD associated with high BMI over a 32-year period and predicted the disease burden trend over the next 29 years. Compared with 1990, ASMR and ASDR caused by T2DM-Associated CKD related to high BMI showed an upward trend in 2021, with increases of 78.4% and 62.7%, respectively (Table 1). There are significant geographical differences: the ASMR and ASDR in Andean Latin America is 6.9/100,000 and 147.4/100,000, the highest in the world; while Eastern Europe is the lowest, at 0.51/100,000. With a sound medical system and high disease diagnosis rates, high-income regions can detect and inter-vene earlier, thereby effectively reducing the disease burden [15,16] In contrast, medical resources are relatively scarce in low- and middle-income areas. The westernization of diets and reduced physical activity brought about by economic development in many developing countries have further contributed to the prevalence of overweight/obesity and diseases related to high BMI [17]. The prevailing diet in Andean Latin America is dominated by high-calorie, low-nutrient-density foods, and diabetes mellitus and T2DM-Associated CKD have a high prevalence and are often undertreated in patients. DN remains one of the leading causes of ESRD, accounting for approximately 36% of CKD cases. In the states of Jalisco and Aguascalientes, the prevalence of DN reaches as high as 48% [18]. In high-income Latin American countries such as Argentina and Brazil, dialysis coverage is relatively high. By contrast, in low-income countries like Nicaragua, coverage remains limited. Overall, dialysis availability shows a positive correlation with both GDP and life expectancy. A study of Mexican patients with kidney failure revealed that most patients did not receive specialized kidney care, and that patients' renal function was very poor at the time of initiation of dialysis, leading to a high post-dialysis mortality rate. Mexican

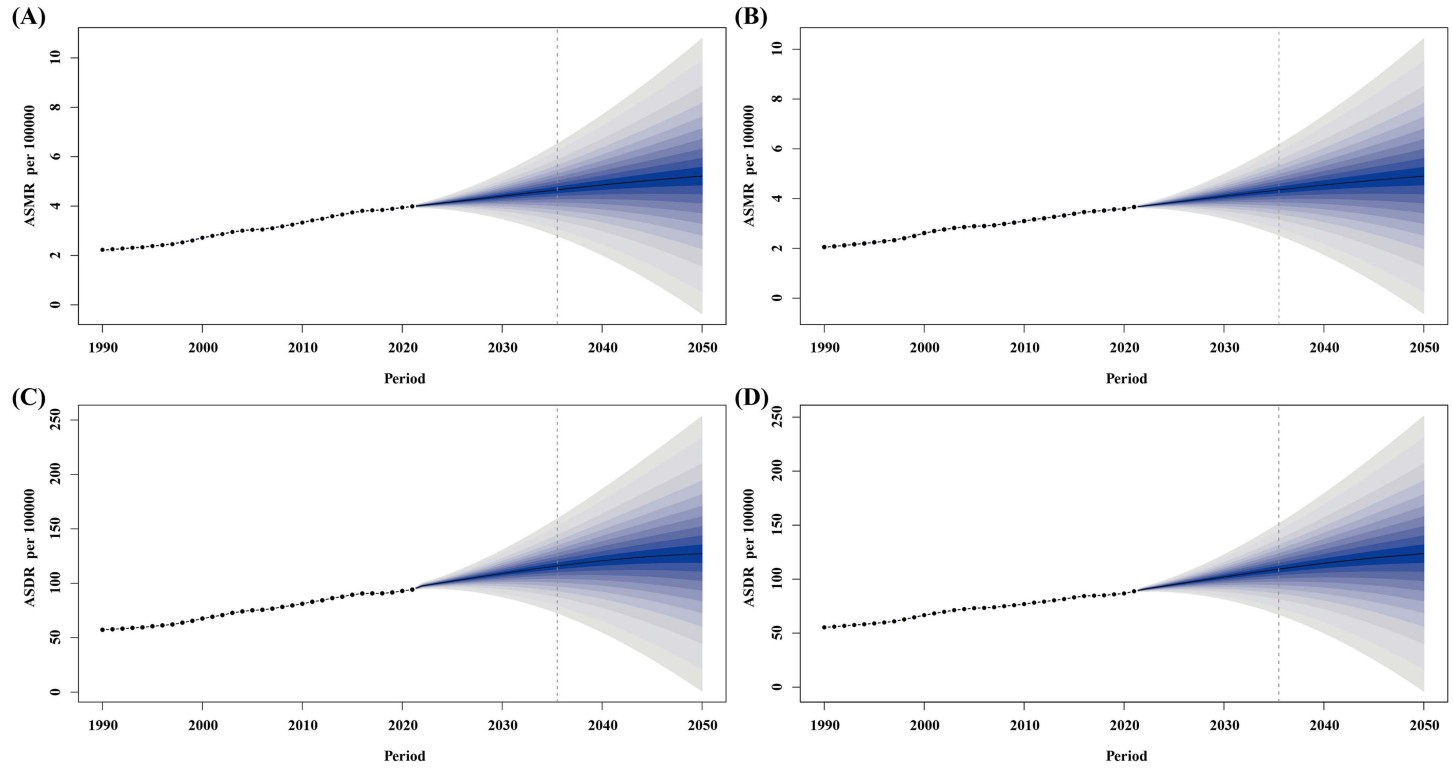

**Fig 6. Global trends in the burden of T2DM-Associated CKD related to high BMI, 2022-2050. (A)** ASMR for males. **(B)** ASMR for females. **(C)** ASDR for males. **(D)** ASDR for females. BMI, body mass index; ASMR, Age-standardized mortality rate; ASDR, Age-standardized disability-adjusted life year rate.

patients had almost three times the risk of death compared to patients in the United States [19]. Additionally, in Latin America, particularly in agricultural regions, long-term environmental factors such as heat stress and pesticide exposure significantly increase the risk of CKD. For example, Guatemala and Honduras have reported CKD cases attributed to these factors, and CKD patients in these regions are often young men, further exacerbating the public health burden [18]. Through region-specific evidence, we can see that common risk factors such as overweight/obesity and diabetes have differing health impacts across regions, closely tied to socio-economic conditions, cultural backgrounds and accessibility of health services. Based on these differences, it becomes particularly important to develop region-specific public health interventions. For example, in high-income countries, interventions may focus more on improving dietary composition and promoting exercise, whereas in low-income countries, increasing access to early screening and treatment of diabetes is more urgent. (Fig 1).

High BMI significantly increases the risk of developing T2DM-Associated CKD through a variety of biological mechanisms, including insulin resistance, chronic inflammatory response and high glomerular filtration [20,21]. First, insulin resistance is a core metabolic feature of individuals with high BMI, which leads to lipid metabolic disorders and elevated levels of free fatty acids (FFAs). FFAs impair glomerular function by activating inflammatory signaling pathways. Their derivatives, such as diacylglycerol (DAG) and ceramide, inhibit key steps in the insulin signaling pathway, including the phosphorylation of insulin receptor substrates, thereby exacerbating insulin resistance and kidney damage [22,23]. Second, chronic inflammatory response is another significant mechanism. The adipose tissue of individuals with high BMI is excessively expanded, which can secrete pro-inflammatory cytokines (such as TNF-α, IL-6 and IL-1β), induce

oxidative stress by activating signal pathways such as NF-κB, thereby damaging glomeruli and renal tubular cells [24,25]. In addition, these pro-inflammatory factors can attract macrophages and T cells into the renal tissue, exacerbating the local inflammatory response and leading to glomerulosclerosis [24,26]. Finally, a high filtration state is an important feature of high BMI-related T2DM-Associated CKD. High BMI leads to increased intra-glomerular pressure and hemodynamic abnormalities. Long-term high filtration state promotes thickening and sclerosis of the glomerular basement membrane, accelerating the decline of renal function [23,26]. As the overweight/obesity rate continues to rise worldwide, the disease burden of T2DM-Associated CKD may further increase in the future, and there is an urgent need to develop new treatments. A randomized controlled trial involving patients with T2D and CKD showed that semaglutide significantly reduced the risk of major events in kidney disease, including the onset of renal failure and the rate of decline in renal function, by 24% compared with the placebo group (Hazard ratio, 0.76; P = 0.0003) [27]. The future burden of T2DM-Associated CKD may change with the use of these emerging drugs, and the long-term impact needs to be further assessed in follow-up studies.

The burden of T2DM-Associated CKD in low-, medium-, and high-SDI regions exhibits a U-shaped trend, reflecting the complex changes in the disease across different stages of economic development. In high-SDI regions, these indicators show a declining trend, primarily attributed to advancements in medical technology and optimized disease management [16]. The incidence rates of ASMR and ASDR are lowest in low-SDI regions (1.50 and 38.01 per 100,000, respectively), but these rates may be underestimated due to underdiagnosis [17]. In regions with moderate SDI, ASMR and ASDR reached their peak levels (2.46 and 58.59 per 100,000, respectively). Many countries have experienced a shift in disease burden from infectious diseases to chronic non-communicable diseases (such as cardiovascular diseases, diabetes, and cancer) [28]. This shift is often accompanied by changes in socioeconomic structure, Westernization of lifestyles (such as dietary habits and reduced physical activity), and aging [17]. These shifts in disease burden require public health interventions to adapt during the transition process. Overall, the impact of economic development on the disease burden is dual in nature. On one hand, improved medical resources and strengthened public health policies have significantly reduced the burden of disease. On the other hand, lifestyle changes and increased rates of overweight/obesity during the early stages of economic development may lead to an increased disease burden in the short term [29]. These trends reveal the profound impact of socioeconomic development, medical resource allocation and lifestyle changes on the global burden of disease, providing an important reference for optimizing public health policies.

There are significant differences between the sexes in T2DM-Associated CKD related to high BMI (Fig 5). The number of deaths due to T2DM-Associated CKD caused by high BMI is higher in women than in men, but the ASMR in men is higher than in women, which reflects the higher mortality risk in men of the same age. This phenomenon may be closely related to the interplay of biological and social behavioral factors. Men are more prone to accumulating visceral fat than women, and the accumulation of visceral fat is closely associated with metabolic issues related to diabetes [30]. Additionally, men have lower metabolic tolerance, which may lead to a higher risk of complications in cases of high BMI. Estrogen plays a certain metabolic protective role in women, helping them better regulate body fat distribution, particularly abdominal fat, thereby reducing the risk of developing T2DM-Associated CKD [31]. In addition to biological factors, social behavioral factors also play an important role in gender differences. Men are generally more likely than women to delay seeking medical care, leading to a lack of early diagnosis and intervention, which in turn exacerbates the condition and increases the risk of mortality. Meanwhile, women have stronger awareness of health management and self-care, which helps them identify diseases earlier and receive effective treatment [32]. These behavioral differences further exacerbate gender disparities in ASMR. Notably, the gap in T2DM-Associated CKD-related disease burden between men and women has widened over time. In recent years, gender differences in overweight/obesity rates and the rising overweight/obesity rate among men may be key factors contributing to this widening gap [33]. Gender differences in overweight/obesity are closely associated with socioeconomic factors, lifestyle factors (such as the Westernization of dietary habits), and work-related stress, which may be more pronounced in men.

Potential interventions and policy options cover a number of areas. First, through health education and behavioral change campaigns, people are encouraged to adopt healthier lifestyles, such as eating a balanced diet and increasing physical activity. Australia has launched a program called "Heart Foundation Walking" aimed at promoting physical activity among residents through community walking. The success of the program has not only improved the health of participants but also reduced rates of cardiovascular disease and obesity [34]. Secondly, enhancing early screening and diagnosis of diseases and promoting early detection and intervention, especially regular medical check-ups for high-risk groups. Public health facilities and telemedicine services are being strengthened. Third, promoting health promotion measures through legislation and policies, such as improving food labelling, promoting salt and sugar control policies, and implementing a sugar tax. Mexico implemented a tax policy on sugary drinks in 2014, reducing consumption by 6% and alleviating obesity to some extent [35]. In 2018, the Philippines implemented a 20% tax on sugary beverages, projected to prevent approximately 2,775 deaths over 20 years. This measure is expected to reduce 13,632 cases of ischemic heart disease and new cardiac events, 5,287 ischemic strokes, and 21,763 cases of T2D [36]. In addition, reducing poverty and improving education and employment opportunities through socio-economic policies can also help to improve health fundamentally.

This study has some limitations. Firstly, although the study utilized the GBD database, which covers 204 countries and territories, there are limitations regarding the completeness and reliability of data for some low-income countries and territories. While the GBD data offer estimates of the prevalence and burden of T2DM-associated CKD, they do not include specific metrics such as eGFR, KDIGO stages, or the urinary albumin-to-creatinine ratio, all of which are essential for staging CKD. Both of these may have implications for the generalizability and accuracy of the study's findings. Secondly, the BAPC model assumes that past trends will continue, but public health interventions, new drug therapies (such as GLP-1RAs and SGLT2i), and socioeconomic changes may have a significant impact on these predictions. Therefore, the BAPC model is not intended for short-term policy forecasting but rather to illustrate potential trend trajectories under the assumption of the current status quo. However, the BAPC model relies on historical data for trend forecasting and is sensitive to sudden events or nonlinear changes (such as public health interventions and economic fluctuations), which may affect the accuracy of future trend forecasts. To enhance the model's responsiveness, future research could incorporate data on external factors (such as public health interventions and economic fluctuations) and adopt nonlinear modeling methods or system dynamics models to better capture nonlinear changes in health trends. Thirdly, it is also worth noting that because the GBD model uses data based on the Geisinger Health System for the derivation of cause distributions, and in particular because the model applies data from the United States to the rest of the world, these data should not be used as specific country-level projections. Instead, they are primarily used to reveal changes in burden trends between global and regional levels. Future research should endeavor to improve these limitations by using more representative and region-specific health data to improve the accuracy and reliability of projections. In addition, this study did not fully incorporate potential factors such as dietary culture differences and genetic backgrounds, which may significantly affect BMI levels and the risk of related diseases in different regions and populations. Therefore, future studies should expand the scope of data coverage, especially strengthen data collection and quality control in low-income areas; and integrate multi-dimensional factors such as dietary culture and genetic backgrounds to more comprehensively assess the disease burden of T2DM-Associated CKD caused by high BMI.

## Supporting information

**S1 Table. ASMR of T2DM-Associated CKD associated with high BMI in 204 countries and regions in 2021.** (XLSX)

**S2 Table. ASDR of T2DM-Associated CKD associated with high BMI in 204 countries and regions in 2021.** (XLSX)

## Acknowledgments

We sincerely thank the Global Burden of Disease research team for providing the publicly available data that formed the basis of the study, and the co-authors for their invaluable support in data collection, analyses, and writing of the paper.

## Author contributions

**Conceptualization:** Jing Zhang.

**Data curation:** Jing Zhang.

**Formal analysis:** Zhen Sun.

**Investigation:** Wenxuan Li, Ke Si, Yajing Huang.

**Methodology:** Wenxuan Li, Ke Si, Yajing Huang.

**Project administration:** Yunyang Wang, Yu Xue, Wenshan Lv.

**Resources:** Yangang Wang.

**Software:** Wenxuan Li, Ke Si, Yajing Huang.

**Supervision:** Yangang Wang.

**Validation:** Yunyang Wang, Yu Xue, Wenshan Lv.

**Visualization:** Zhen Sun.

**Writing – original draft:** Jing Zhang.

**Writing – review & editing:** Lili Xu.

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
