## [Decision Letter · Decision Letter 0]

17 Aug 2025

PONE-D-25-38032Trends and Projections of the Global Burden of Type 2 Diabetic Nephropathy Related to High BMI: A Global Burden of Disease Study 2021PLOS ONE

Dear Dr. wang,

Thank you for submitting your manuscript to PLOS ONE. After careful consideration, we feel that it has merit but does not fully meet PLOS ONE’s publication criteria as it currently stands. Therefore, we invite you to submit a revised version of the manuscript that addresses the points raised during the review process.

**ACADEMIC EDITOR: **

Thank you for submitting your manuscript to PLOS ONE. The peer-review process has been completed. The reviewers have requested some major issues that need to be addressed before the manuscript can be considered for publication. The detailed feedback from reviewers is included below/attached for your reference.

We kindly request that you address these points in your revised manuscript and provide a response letter detailing the changes made. Please submit the revised version of your manuscript along with the response letter through our submission system.

We look forward to receiving your revised manuscript.

Kind regards,

Claudio Alberto Dávila-Cervantes, Ph.D.

Academic Editor

PLOS ONE

Journal Requirements:

2. We note that Figure 2 in your submission contain [map/satellite] images which may be copyrighted. All PLOS content is published under the Creative Commons Attribution License (CC BY 4.0), which means that the manuscript, images, and Supporting Information files will be freely available online, and any third party is permitted to access, download, copy, distribute, and use these materials in any way, even commercially, with proper attribution. For these reasons, we cannot publish previously copyrighted maps or satellite images created using proprietary data, such as Google software (Google Maps, Street View, and Earth). For more information, see our copyright guidelines: http://journals.plos.org/plosone/s/licenses-and-copyright.

3. We note you have included a table to which you do not refer in the text of your manuscript. Please ensure that you refer to Table 3 and 4 in your text; if accepted, production will need this reference to link the reader to the Table.

4. Please include a new copy of Table 1, 2, 3, and 4 in your manuscript; the current table is difficult to read. Please follow the link for more information: https://blogs.plos.org/plos/2019/06/looking-good-tips-for-creating-your-plos-figures-graphics/"

6. Please remove all personal information, ensure that the data shared are in accordance with participant consent, and re-upload a fully anonymized data set.

Reviewers' comments:

Reviewer's Responses to Questions

**Comments to the Author**

1. Is the manuscript technically sound, and do the data support the conclusions?

Reviewer #1: Yes

Reviewer #2: Partly

2. Has the statistical analysis been performed appropriately and rigorously? 

Reviewer #1: Yes

Reviewer #2: Yes

3. Have the authors made all data underlying the findings in their manuscript fully available?

Reviewer #1: Yes

Reviewer #2: Yes

4. Is the manuscript presented in an intelligible fashion and written in standard English?

Reviewer #1: Yes

Reviewer #2: No

5. Review Comments to the Author

Reviewer #1: The manuscript titled “Trends and Projections of the Global Burden of Type 2 Diabetic Nephropathy Related to High BMI: A Global Burden of Disease Study 2021” addresses an important and timely public health issue, using comprehensive GBD 2021 data to describe trends and make projections. The topic is relevant, the scope is global, and the statistical approaches (Joinpoint regression, decomposition analysis, Bayesian Age-Period-Cohort modeling) are appropriate. The manuscript is generally well-structured and clear, with a logical flow from introduction to discussion.

However, there are some methodological, interpretative, and presentation issues that should be addressed before the manuscript can be considered for publication.

Major Comments

1. Clarity on Data Sources and GBD Versions

The manuscript alternately refers to GBD 2021 and GBD 2024 data. This is confusing and should be clarified to avoid inconsistencies. If GBD 2024 updates were used, the title, abstract, and methods should reflect that clearly.

The authors should specify whether they used raw GBD estimates or age-standardized, modeled data from IHME, and how they handled missing or low-certainty data for some regions.

2. Definition of “High BMI”

The definition (>20–23 kg/m² depending on region) is unconventional, as most obesity-related research uses ≥25 kg/m² for overweight and ≥30 kg/m² for obesity in adults. This threshold should be justified, with clear reference to GBD’s operational definition and why it differs from WHO’s cut-offs.

3. Interpretation of Regional Trends

The discussion sometimes attributes differences solely to dietary habits or healthcare access without citing region-specific studies. While plausible, these interpretations would be stronger with region-specific epidemiological evidence.

The SDI-related trends (e.g., U-shaped relationships in low-to-middle SDI settings) deserve deeper exploration, potentially with supporting literature on health transition stages.

4. Projection Model Limitations

The Bayesian Age-Period-Cohort model assumes continuation of past trends. The manuscript should emphasize that public health interventions, new pharmacotherapies (e.g., GLP-1 receptor agonists, SGLT2 inhibitors), and socio-economic changes could substantially alter these projections.

The text should note explicitly that the model is not designed for short-term policy forecasting but for illustrating possible trajectories under status quo assumptions.

5. Sex Differences

The manuscript notes that men have higher ASMR/ASDR despite fewer absolute deaths than women. The discussion could benefit from integrating biological (e.g., fat distribution, hormonal effects) and socio-behavioral explanations, with literature support.

The widening gap between sexes over time should be contextualized with trends in obesity prevalence by sex.

6. Policy Implications

While the discussion suggests general interventions (diet, exercise, taxation), these remain broad. The paper could be strengthened by including specific examples of successful programs in high-burden regions, linked to the observed epidemiological patterns.

Minor Comments

1. Language and Style

The English is generally clear, but some sentences are overly long and would benefit from simplification for readability.

Certain terms should be standardized (e.g., “high body mass index” vs. “high BMI”).

2. Figures and Tables

Figure legends should be fully self-contained, describing abbreviations and clarifying that rates are age-standardized.

Consider including a supplementary table with country-level ASMR/ASDR values for transparency.

3. Referencing

Some statements in the discussion lack direct citations, especially those linking trends to dietary patterns or healthcare access.

References should be checked for format consistency with the journal’s style.

4. Ethics Statement

Although GBD data are publicly available, the ethics statement could note explicitly that no human participants were directly involved.

Recommendation:

Major Revision – The study is valuable and relevant, but revisions are needed to clarify definitions, ensure methodological transparency, strengthen the interpretation of results, and align discussion points more closely with evidence.

Reviewer #2: Dear Academic Editor,

Please find below my consolidated reviewer comments for the manuscript entitled “Trends and Projections of the Global Burden of Type 2 Diabetic Nephropathy Related to High BMI: A Global Burden of Disease Study 2021”.

This research manuscript explores the global burden of type 2 diabetic nephropathy (T2DN) associated with high body mass index (BMI), using data from the Global Burden of Disease (GBD) Study 2021. The study analyzes historical trends from 1990 to 2021 and projects scenarios through 2050, examining mortality and disability-adjusted life years (DALYs). It highlights that high BMI is an important driver of increasing T2DN, with notable regional and sex disparities. The research emphasizes the urgency of interventions to prevent obesity and to improve access to healthcare.

I would recommend a Major Revision focused on clarifying the manuscript’s novel contribution, strengthening the methodological justification, and improving precision in terminology (chronic kidney disease due to type 2 diabetes) and the BMI cut-points used. Implementing the suggested revisions would increase the manuscript’s clarity, reproducibility, and usefulness for policy-makers.

Context and novelty relative to recent studies

Why: Several recent GBD-based analyses address the burden of chronic kidney disease (CKD) attributable to type 2 diabetes and the role of high BMI (e.g., Wang et al., 2025 and others). Clarifying what is new in this manuscript will help readers and editors assess its added value.

Suggestion: The authors could add a brief paragraph in the Introduction that cites these recent studies and explains what the present manuscript adds (for example: projection horizon extended to 2050; an alternative projection method; more granular subregional/age/sex stratification; additional sensitivity analyses).

Clarify and justify the definition of “high BMI”

Why: Using thresholds such as 20–23 kg/m² labeled as “high BMI” can be confused with international categories for overweight/obesity (≥25/≥30 kg/m²). Interpreting results as “obesity-driven” without clarifying this distinction can be misleading.

Suggestion: The authors could clarify the exact source of these thresholds (e.g., GBD documentation, WHO Expert Consultation for Asian populations, or another reference), indicate which countries/regions apply which threshold (if applicable), and — where appropriate — use the term “elevated/high BMI” in Results and Conclusions instead of “obesity.”

Consistent terminology tied to the GBD definition (T2DM-Associated CKD / DKD)

Why: The term “diabetic nephropathy” usually implies clinical phenotypes (albuminuria, eGFR, KDIGO stages) that the GBD attributional category does not permit distinguishing. Using a neutral, explicit label avoids overinterpretation and is consistent with terminology adopted in other GBD-based works.

Suggestion: The authors could define and consistently use “T2DM-Associated CKD”, “CKD-T2DM” (or alternatively “Diabetic Kidney Disease, DKD” when used as a broad, non-phenotypic term). As noted in their Limitations (GBD does not distinguish proteinuria nor KDIGO stages), the Methods should state this explicitly.

Methodological details for reproducibility (Joinpoint, BAPC, code)

Why: Small modeling details influence replicability and the interpretation of projections.

Suggestion: The authors could provide methodological details in Methods: how data were imported into Joinpoint (R package or script), the maximum number of joinpoints allowed, the number of permutations used for permutation tests, significance thresholds, BAPC parameters (priors, MCMC iterations), software and package versions. If feasible, deposit analysis scripts in a public repository (e.g., GitHub, Zenodo) and provide the link.

Make the Data Availability Statement explicit

Why: Citing the GBD portal is helpful, but specifying the exact outputs improves reproducibility.

Suggestion: The authors could list the exact outputs extracted (deaths, DALYs, ASMR, ASDR), years, age groups, the cause definition used, and file formats. Providing the extracted dataset (CSV) as supplementary material would be useful.

Addressing these points would significantly improve the manuscript’s methodological transparency, terminological precision, and compliance with PLOS ONE’s style and reporting expectations.

Thank you for the opportunity to review this work.

6. PLOS authors have the option to publish the peer review history of their article (what does this mean? ). If published, this will include your full peer review and any attached files.

**Do you want your identity to be public for this peer review?** For information about this choice, including consent withdrawal, please see our Privacy Policy .

Reviewer #1: No

Reviewer #2: No

---

## [Author Response · Author response to Decision Letter 1]

4 Sep 2025

Thank you for your constructive comments on our manuscript entitled ‘Trends and Projections of the Global Burden of Type 2 Diabetic Nephropathy Related to High BMI: A Global Burden of Disease Study 2021’ (ID: PONE-D-25-38032). We appreciate the opportunity to revise and improve our manuscript. We have carefully addressed all of your comments and have made the necessary changes to the manuscript. Following this, we have responded to your comments point by point.

To the academic editor:

Comment1: Please ensure that your manuscript meets PLOS ONE's style requirements, including those for file naming.

Response: Thank you for your reminder. We have made the revisions according to the PLOS ONE format template you provided.

Comment2: We note that Figure 2 in your submission contain [map/satellite] images which may be copyrighted.

Response: The issue you raised is very valuable. We have removed Figure 2 and provided the relevant data as Supporting Information.

Comment3: We note you have included a table to which you do not refer in the text of your manuscript. Please ensure that you refer to Table 3 and 4 in your text.

Response: Thank you for your suggestion. Our article does not contain Tables 3 and 4. We have conducted a thorough review to ensure that such errors will not occur again.

Comment4: Please include a new copy of Table 1, 2, 3, and 4 in your manuscript; the current table is difficult to read.

Response: We have added new versions of Tables 1 and 2 to the manuscript (pages 8–9).

Comment5: Please include captions for your Supporting Information files at the end of your manuscript, and update any in-text citations to match accordingly.

Response: We have listed the titles of the Supporting Information files at the end of the paper and verified the citations in the main text (page 27, lines 527–531).

Comment6: Please remove all personal information, ensure that the data shared are in accordance with participant consent, and re-upload a fully anonymized data set.

Response: Our research is based on the GBD database, whose data is publicly available and does not contain personal information, only aggregated regional data. We have uploaded a new data set.

Comment7: If the reviewer comments include a recommendation to cite specific previously published works, please review and evaluate these publications to determine whether they are relevant and should be cited. There is no requirement to cite these works unless the editor has indicated otherwise.

Response: We thank the editor for the guidance regarding citation of reviewer-recommended references. We carefully examined the suggested works and, in accordance with the editorial instructions, did not cite them solely on the basis of reviewer recommendation. Instead, we selected references strictly based on their direct relevance to the content and arguments of our manuscript. We hope this approach aligns with the editorial policy.

To the reviewer 1

Major Comment1: Clarity on Data Sources and GBD Versions. The manuscript alternately refers to GBD 2021 and GBD 2024 data. This is confusing and should be clarified to avoid inconsistencies. If GBD 2024 updates were used, the title, abstract, and methods should reflect that clearly. The authors should specify whether they used raw GBD estimates or age-standardized, modeled data from IHME, and how they handled missing or low-certainty data for some regions.

Response: We sincerely thank the reviewer for the careful reading and constructive comments. The mention of “GBD 2024” in the manuscript was an unintentional error on our part, and we have corrected it to “GBD 2021” throughout the text (page 4, line 69). We apologize for this oversight.

As clarified, our analysis was based on the age-standardized, modeled estimates provided by the Institute for Health Metrics and Evaluation (IHME), rather than raw GBD data. Regarding data quality, we conducted a careful data-cleaning process. Specifically, we removed duplicate entries and ensured internal consistency across indicators. For regions with missing or low-certainty estimates, we followed the IHME protocol of using modeled age-standardized estimates, which incorporate statistical adjustments to account for data sparsity. Where IHME provided no reliable estimate for a given region, we excluded these data points from the analysis to avoid introducing bias.

We hope this clarification resolves the reviewer’s concern and improves the transparency of our methodology.

Major Comment2: Definition of “High BMI”. The definition (>20–23 kg/m² depending on region) is unconventional, as most obesity-related research uses ≥25 kg/m² for overweight and ≥30 kg/m² for obesity in adults. This threshold should be justified, with clear reference to GBD’s operational definition and why it differs from WHO’s cut-offs.

Response: We appreciate the valuable comments provided by the reviewers. After reviewing relevant literature in the field, we found that most studies adopt BMI ≥ 25 kg/m² as the criterion for overweight and ≥ 30 kg/m² as the criterion for obesity. Upon consulting the GBD database, we observed that the original data utilized a BMI range of >20–23 kg/m². It is important to note that the publicly available GBD data has been cleaned and standardized to a BMI threshold of ≥25 kg/m². Based on this, we have made corresponding revisions to the manuscript (page 5, line 87). We are grateful for the reviewer's reminder.

Major Comment3: Interpretation of Regional Trends. The discussion sometimes attributes differences solely to dietary habits or healthcare access without citing region-specific studies. While plausible, these interpretations would be stronger with region-specific epidemiological evidence.

The SDI-related trends (e.g., U-shaped relationships in low-to-middle SDI settings) deserve deeper exploration, potentially with supporting literature on health transition stages.

Response: We sincerely appreciate the reviewers' valuable feedback on our paper, particularly the suggestions regarding the interpretation of regional trends. Based on your comments, we have incorporated additional region-specific epidemiological data and supporting literature into the first paragraph of the Discussion section (pages 12-13, lines 216-218, 220-225, 227-232). Specifically, we cited data from the 2019 Latin American Dialysis and Renal Transplant Registry (LADRTR) to detail variations in diabetic nephropathy prevalence across different countries and regions. Additionally, we discussed non-traditional CKD cases in parts of Latin America linked to agricultural environmental factors, particularly emphasizing the impact of heat stress and pesticide exposure on disease occurrence. These modifications enhance the regional specificity of our discussion and provide more targeted epidemiological evidence.

Regarding your suggestion about discussing the U-shaped relationship in low-to-medium SDI regions, we fully agree with your perspective and have further explored this trend in the article. In the revised manuscript, we incorporated the U-shaped pattern of changes in the burden of diabetes-related kidney disease across low, medium, and high SDI regions, highlighting the complex influence of economic development stages on disease burden. Specifically, we detail how advances in medical technology and optimized disease management significantly reduce the burden of diabetic kidney disease in high-SDI regions. Conversely, in medium-SDI regions, the transition of disease burden from infectious to chronic non-communicable diseases, coupled with Westernized lifestyles and population aging, exacerbates the disease burden (Page 13, lines 236-241). Furthermore, we cite relevant literature to further substantiate these interpretations of the changes and highlight the importance of this trend for optimizing public health policies.

Once again, we appreciate your meticulous review of our article and believe these revisions enhance its comprehensiveness and depth.

Major Comment4: Projection Model Limitations. The Bayesian Age-Period-Cohort model assumes continuation of past trends. The manuscript should emphasize that public health interventions, new pharmacotherapies (e.g., GLP-1 receptor agonists, SGLT2 inhibitors), and socio-economic changes could substantially alter these projections.

The text should note explicitly that the model is not designed for short-term policy forecasting but for illustrating possible trajectories under status quo assumptions.

Response: Thank you for your reminder. We fully recognize the limitations of the BAPC model in assuming the continuation of historical trends and acknowledge that public health interventions, new drug therapies, and socioeconomic changes may profoundly impact future trends.

Following your suggestion, we have further emphasized this point in the revised manuscript, explicitly stating that the BAPC model is not designed for short-term policy forecasting but rather to illustrate potential trend trajectories under the status quo assumption. We specifically added that the BAPC model projects trends based on historical data and thus cannot fully capture the impact of sudden events or nonlinear changes—such as public health interventions and economic fluctuations—on future trends. To enhance the model's resilience, future research could incorporate external factors like public health interventions and economic fluctuations, and adopt nonlinear modeling approaches or system dynamics models to better capture nonlinear shifts in health trends (page 17-18, lines 332-341).

We sincerely thank you again for your insightful contributions to our work. Your recommendations have been instrumental in improving this manuscript.

Major Comment5: Sex Differences. The manuscript notes that men have higher ASMR/ASDR despite fewer absolute deaths than women. The discussion could benefit from integrating biological (e.g., fat distribution, hormonal effects) and socio-behavioral explanations, with literature support.

The widening gap between sexes over time should be contextualized with trends in obesity prevalence by sex.

Response: Thank you for your meticulous review and valuable suggestions regarding my paper. I have made the relevant revisions based on your feedback.

Firstly, regarding the relationship between gender differences and ASMR/ASDR, you suggested incorporating biological and sociological explanations. In my revisions, I have supplemented the biological factors. Specifically, I noted that males are more prone to accumulating visceral fat, which is closely associated with diabetes and metabolic disorders (Page 15, Lines 293-296). Additionally, males exhibit relatively lower metabolic tolerance, potentially leading to heightened complication risks under high BMI conditions. In contrast, female estrogen exerts protective effects, particularly in body fat distribution, by regulating abdominal fat and thereby reducing T2DN risk (Page 16, Lines 297-299). Relevant literature supporting this section has also been incorporated.

Secondly, regarding the relationship between gender differences and obesity rate trends, I highlighted that the recent rise in male obesity rates may be a key factor exacerbating gender disparities in the disease burden associated with T2DN. Gender disparities in obesity are closely linked to socioeconomic factors, lifestyle factors (such as the prevalence of Western dietary habits), and work-related stress, particularly pronounced among men (Page 16, lines 308-310).Once again, thank you for your valuable suggestions. Your feedback has significantly enhanced the quality of this paper. Should you have any further recommendations or questions, please do not hesitate to share them. I would be delighted to continue refining this work.

Major Comment6: Policy Implications. While the discussion suggests general interventions (diet, exercise, taxation), these remain broad. The paper could be strengthened by including specific examples of successful programs in high-burden regions, linked to the observed epidemiological patterns.

Response: Thank you for your meticulous review and valuable suggestions regarding my paper. Based on your feedback, we have revised the article and incorporated specific success stories to enhance its practicality and relevance.

Specifically, we have added the case study of Mexico's 2014 sugar-sweetened beverage tax policy. Research indicates this policy reduced consumption of sugary drinks by 6% and effectively mitigated obesity issues (Page 17, Lines 321-322). Additionally, we supplemented the article with the successful case of Australia's “Heart Foundation Walk Program.” This initiative promoted physical activity among residents through community walking events, significantly improving participants' health levels while reducing cardiovascular disease and obesity rates (Page 16, Lines 313-316).

We believe these concrete examples effectively support the interventions discussed in the paper. Combined with the epidemiological models, they make the discussion section more specific and practical. Thank you once again for your valuable suggestions, which have greatly enhanced the article's quality. Should you have any further recommendations or questions, please do not hesitate to inform me, and I will continue to refine the manuscript.

Minor Comment1: Language and Style. The English is generally clear, but some sentences are overly long and would benefit from simplification for readability. Certain terms should be standardized (e.g., “high body mass index” vs. “high BMI”).

Response: We appreciate the reviewers' valuable feedback. We have revised the manuscript according to their suggestions, streamlined certain sentences to enhance readability, and standardized the use of terminology.

Minor Comment2: Figure legends should be fully self-contained, describing abbreviations and clarifying that rates are age-standardized.

Consider including a supplementary table with country-level ASMR/ASDR values for transparency.

Response: We appreciate the reviewer's suggestions. We have revised the figure and table captions to ensure they are self-contained and clearly describe the abbreviations, explicitly noting that all ratios are age-standardized. Additionally, we have submitted supplementary tables containing more detailed data to enhance transparency.

Minor Comment3: Some statements in the discussion lack direct citations, especially those linking trends to dietary patterns or healthcare access. References should be checked for format consistency with the journal’s style.

Response: We appreciate the reviewers' detailed feedback. We have incorporated direct citations and additional case examples into the Discussion section. Furthermore, all citations have been reviewed and standardized according to the journal's formatting requirements.

Minor Comment4: Ethics Statement. Although GBD data are publicly available, the ethics statement could note explicitly that no human participants were directly involved.

Response: We have explicitly stated in the “method” and “Ethics approval and consent to participate” section that the GBD data used in this paper is publicly available and does not directly involve human participants. Thank you; your feedback helps further enhance the rigor of the article.

To the reviewer 2

Comment1: Context and novelty relative to recent studies. Why: Several recent GBD-based analyses address the burden of chronic kidney disease (CKD) attributable to type 2 diabetes and the role of high BMI (e.g., Wang et al., 2025 and others). Clarifying what is new in this manuscript will help readers and editors assess its added value. Suggestion: The authors could add a brief paragraph in the Introduction that cites these recent studies and explains what the present manuscript adds (for example: projection horizon extended to 2050; an alternative projection method; more granular subregional/age/sex stratification; additional sensitivity analyses).

Response: Thank you for your valuable feedback on our paper. We have revised the relevant sections of the manuscript as per your suggestions to ensure our research's unique contributions and relationship to prior work are more clearly articulated.

In the Introduction, we cited Tan et al.'s analy

---

## [Decision Letter · Decision Letter 1]

8 Sep 2025

PONE-D-25-38032R1Trends and Projections of the Global Burden of T2DM-Associated CKD Related to High BMI: A Global Burden of Disease Study 2021PLOS ONE

Dear Dr. wang,

Thank you for submitting your manuscript to PLOS ONE. After careful consideration, we feel that it has merit but does not fully meet PLOS ONE’s publication criteria as it currently stands. Therefore, we invite you to submit a revised version of the manuscript that addresses the points raised during the review process.

We look forward to receiving your revised manuscript.

Kind regards,

Claudio Alberto Dávila-Cervantes, Ph.D.

Academic Editor

PLOS ONE

Journal Requirements:

Reviewers' comments:

Reviewer's Responses to Questions

**Comments to the Author**

1. If the authors have adequately addressed your comments raised in a previous round of review and you feel that this manuscript is now acceptable for publication, you may indicate that here to bypass the “Comments to the Author” section, enter your conflict of interest statement in the “Confidential to Editor” section, and submit your "Accept" recommendation.

Reviewer #1: All comments have been addressed

Reviewer #2: All comments have been addressed

2. Is the manuscript technically sound, and do the data support the conclusions?

Reviewer #1: Yes

Reviewer #2: Yes

3. Has the statistical analysis been performed appropriately and rigorously? 

Reviewer #1: Yes

Reviewer #2: Yes

4. Have the authors made all data underlying the findings in their manuscript fully available?

Reviewer #1: Yes

Reviewer #2: Yes

5. Is the manuscript presented in an intelligible fashion and written in standard English?

Reviewer #1: Yes

Reviewer #2: Yes

6. Review Comments to the Author

Reviewer #1: This is a timely and relevant analysis of the global burden of T2DM-associated CKD attributable to high BMI using GBD 2021 data. The projection horizon to 2050 and the stratification by sex, age, and SDI add clear value.

Strengths include comprehensive use of global data, improved terminology (“T2DM-associated CKD”), and region-specific discussion supported by epidemiological evidence.

Minor points for further improvement:

In the abstract/conclusions, highlight regional disparities and clarify that emerging therapies (GLP-1 RAs, SGLT2i) could alter future trajectories.

Ensure consistent use of terminology (“high BMI” vs. “obesity”; harmonize with GBD definitions).

Provide clarity on data availability (whether extracted CSV or repository link will be accessible).

Simplify long sentences to improve readability, and standardize abbreviations in figures/tables.

Add one more policy example from Asia or Africa to balance global representation.

Overall, this is a solid and well-revised manuscript; I recommend minor revisions to improve clarity and consistency.

This revised manuscript represents a substantial improvement and provides a valuable contribution to the literature on the global burden of T2DM-associated CKD attributable to high BMI. The use of GBD 2021 data, projections to 2050, and stratification by sex, age, and SDI add novelty and strengthen its public health relevance.

The authors have addressed the major reviewer concerns appropriately, including clarification of data sources, consistency of terminology, methodological transparency, and region-specific interpretation. The discussion is richer and includes relevant policy examples.

Remaining issues are minor: improving consistency of terminology (“high BMI” vs. “obesity”), ensuring clarity in the abstract/conclusions (particularly regarding regional disparities and the potential impact of emerging therapies), providing a clear data availability statement, and light language/formatting adjustments.

Reviewer #2: All comments have been addressed by the authors; thank you for the thorough revisions and attention to detail.

7. PLOS authors have the option to publish the peer review history of their article (what does this mean? ). If published, this will include your full peer review and any attached files.

**Do you want your identity to be public for this peer review?** For information about this choice, including consent withdrawal, please see our Privacy Policy .

Reviewer #1: **Yes: ** Ana M Cebrián-Cuenca

Reviewer #2: **Yes: ** Juan Rodrigo Gómez-Bernal

---

## [Author Response · Author response to Decision Letter 2]

16 Sep 2025

Thank you for your thoughtful and constructive feedback on our manuscript. We greatly appreciate the time and effort you have put into reviewing our work. Below, we outline how we have addressed the points you raised:

Reviewer 1:

Comment1: In the abstract/conclusions, highlight regional disparities and clarify that emerging therapies (GLP-1 RAs, SGLT2i) could alter future trajectories.

Response: In response to your suggestion, we have incorporated a discussion on emerging therapies, including GLP-1 receptor agonists (RAs) and SGLT2 inhibitors (SGLT2i), in the abstract. We acknowledge the potential impact of these therapies on the future trends of T2DM-associated CKD, particularly in regions with a high disease burden. (page2, line 34-35)

Comment2: Ensure consistent use of terminology (“high BMI” vs. “obesity”; harmonize with GBD definitions).

Response: We truly appreciate your careful attention to the terminology used throughout the manuscript. In response to your comment regarding the consistent use of terminology (“high BMI” vs. “obesity”), we have carefully reviewed the manuscript and made the necessary revisions. Specifically, we have replaced instances of "obesity" with "overweight/obesity" to better align with the BMI thresholds defined in the methods section (BMI ≥ 25 kg/m²). This change ensures consistency and harmonizes the terminology with the Global Burden of Disease (GBD) definitions, as per your suggestion.

We believe this revision improves the clarity and precision of the manuscript. Thank you once again for your valuable input, and we hope the updated manuscript now meets your expectations.

Comment3: Provide clarity on data availability (whether extracted CSV or repository link will be accessible).

Response: We appreciate your suggestion to clarify the access to the data. As noted in the Methods section, we have provided the link to the relevant database. To further enhance the reproducibility of our work, we have also prepared the extracted data in CSV format as supplementary material. However, due to file size limitations, we were unable to upload the CSV file via the submission platform. If needed, we are happy to provide the data through alternative methods, such as via email or a file-sharing service, and will be glad to share it with you or the editorial team at your convenience. We hope this addresses your concern.

Comment4: Simplify long sentences to improve readability, and standardize abbreviations in figures/tables.

Response: Thank you for your valuable feedback. I have carefully considered your suggestions to simplify long sentences and improve readability. In response to your comment, I have revised several sections of the manuscript. The sentences in lines 59, 79-83, 160-164, 234-237, 257-260, 288-292, and 334-339 have been simplified for better clarity and flow. Additionally, I have standardized the abbreviations used in the figures and tables to ensure consistency throughout the manuscript. I believe these revisions significantly enhance the readability of the manuscript. Thank you again for your thoughtful suggestions, which have greatly contributed to improving the quality of the work.

Comment5: Add one more policy example from Asia or Africa to balance global representation.

Response: Thank you for your valuable feedback. Regarding your suggestion to include policy examples from Asia or Africa, we have incorporated the case of the Philippines—which implemented a sugar tax policy in 2018. The revised version adds the following statement: "In 2018, the Philippines implemented a 20% tax on sugary beverages, projected to prevent approximately 2,775 deaths over 20 years. This measure is expected to reduce 13,632 cases of ischemic heart disease and new cardiac events, 5,287 ischemic strokes, and 21,763 cases of T2D." (page17, line 328-331) This addition aims to present a more balanced perspective by incorporating policy examples from Southeast Asia and South Asia, providing broader context for tax measures addressing global health challenges stemming from unhealthy diets.

We sincerely thank the reviewers for their valuable suggestions on this paper. We have carefully revised each piece of feedback to enhance the accuracy and readability of the manuscript. We believe these modifications will significantly improve the quality of the article. Once again, we extend our gratitude for your valuable time and hard work.

---

## [Editor Report · Decision Letter 2]

17 Sep 2025

Trends and Projections of the Global Burden of T2DM-Associated CKD Related to High BMI: A Global Burden of Disease Study 2021

PONE-D-25-38032R2

Dear Dr. wang,

We’re pleased to inform you that your manuscript has been judged scientifically suitable for publication and will be formally accepted for publication once it meets all outstanding technical requirements.

Kind regards,

Claudio Alberto Dávila-Cervantes, Ph.D.

Academic Editor

PLOS ONE
---

## [Editor Report · Acceptance letter]

PONE-D-25-38032R2

PLOS ONE

Dear Dr. Wang,

I'm pleased to inform you that your manuscript has been deemed suitable for publication in PLOS ONE. Congratulations! Your manuscript is now being handed over to our production team.

Kind regards,

on behalf of

Mr. Claudio Alberto Dávila-Cervantes

Academic Editor

PLOS ONE